# Brown Seaweed Consumption as a Promising Strategy for Blood Glucose Management: A Comprehensive Meta-Analysis

**DOI:** 10.3390/nu15234987

**Published:** 2023-12-01

**Authors:** Yu Rim Kim, Min Ju Park, Soo-yeon Park, Ji Yeon Kim

**Affiliations:** Department of Food Science and Biotechnology, Seoul National University of Science and Technology, 232, Gongneung-ro, Nowon-gu, Seoul 01811, Republic of Korea; 22512167@seoultech.ac.kr (Y.R.K.); alswn1895@seoultech.ac.kr (M.J.P.); sooyeon.park@seoultech.ac.kr (S.-y.P.)

**Keywords:** brown seaweed, algae, blood glucose, meta-analysis

## Abstract

Diabetes is a chronic condition that can lead to various complications; therefore, there is a need to emphasize prevention and management. Dietary interventions, such as the Mediterranean diet or calorie-restricted regimens, coupled with exercise-induced weight reduction, have been recommended for enhancing diabetes management. Seaweeds contain various functional components, such as polyphenols and fucoidan, which have been reported to exert multiple benefits, including blood glucose regulation, improved intestinal health, and enhanced of lipid profiles. The association between blood glucose and seaweed consumption has been established in previous research. We searched the PubMed, RISS, Google Scholar, ScienceDirect, and Cochrane Library databases to identify relevant studies after applying the selection/exclusion criteria, and 23 studies were ultimately included in this analysis. Comprehensive Meta-Analysis (CMA) software version 4.0 was used to assess statistical significance and heterogeneity. In this meta-analysis, postprandial blood glucose, glycated hemoglobin (HbA1c), and Homeostatic Model Assessment of Insulin Resistance (HOMA-IR) levels demonstrated significant improvements in the seaweed group compared to the control group. Conversely, fasting blood glucose and insulin levels did not show significant associations with seaweed consumption. Subgroup analysis revealed that a high dose (1000 mg or more) was more beneficial than a low dose, and seaweeds such as *Laminaria digitata*, *Undaria pinnatifida*, *Acophyllum nodosum*, and *Fucus vesiculosus* were found to be more effective at improving blood glucose levels than control treatments. Therefore, based on our research, seaweed supplementation appears to be a promising strategy for reducing postprandial blood glucose, HbA1c, and HOMA-IR levels, thereby enabling better blood glucose management and leading to a decreased risk of type 2 diabetes.

## 1. Introduction

The shifts in contemporary lifestyles have led to a surge in chronic diseases, with factors such as high-calorie diets and sedentary behavior being major contributors [1,2]. It is widely recognized that approximately 90% of type 2 diabetes cases can be attributed to overweight and obesity [3]. Diabetes is associated with a range of complications, including cardiovascular ailments (e.g., coronary artery disease and heart failure), renal disorders, and cerebrovascular events (e.g., stroke) [4]. Treatment for type 2 diabetes typically involves the administration of oral medications such as insulin, various sulfonylureas, and metformin [5]. Diabetes management can be enhanced using dietary interventions such as the Mediterranean diet or calorie-restricted regimens coupled with exercise-induced weight reduction. Diabetes medications have been reported to be associated with potential side effects [6]. Approximately 30% of individuals using metformin experience gastrointestinal disturbances and vitamin deficiencies, such as vitamin B12 and folic acid deficiency. Additionally, insulin can cause side effects such as weight gain or loss, hypokalemia, and allergic reactions [7]. Consequently, when it comes to medications to treat diabetes, as well as preventive and management strategies, it is often recommended to prioritize prevention through a balanced diet and regulating blood sugar with a low-carbohydrate diet [8,9].

The number of species of algae in the ocean is estimated to range from 30,000 to more than 1 million. Algae are classified into three groups based on their colors: brown algae, green algae, and red algae [10,11]. Throughout history, seaweed consumption has been prevalent, especially in Asia. There are various methods of consuming seaweed [12,13]. It can be ingested as a side dish or in the form of fermented foods [14]. Additionally, seaweed is utilized as a food additive to enhance culinary properties [15]. Moreover, it is consumed in the form of supplements that contain extracts or powders aimed at maintaining and improving health. In recent years, seaweed consumption has spread to Western countries [13]. Seaweed not only contains a high level of protein and dietary fiber as nutritional supplements in foods, but is also rich in polyphenols, polysaccharides, and carotenoids, which are known as health functional constituents [16,17]. InSea2^®^ (InnoVactiv Inc., Rimouski, QC, Canada) is one such dietary supplement, composed of extracts from the brown seaweeds *Ascophyllum nodosum* and *Fucus vesiculosus*, which contain phlorotannins [18]. It has been approved by Health Canada for its two primary benefits: helping maintain healthy blood glucose levels and reducing the glycemic index of ingested foods (NPN 80033840). Polyphenols have been reported to exhibit protective effects against hyperglycemia and oxidative stress [19]. Polysaccharides, such as alginate and fucoidan, have been suggested to assist in increased cholesterol excretion through gel formation within the intestine [20,21]. Numerous in vitro and in vivo studies have validated the antidiabetic mechanisms of seaweed, including the inhibition of α-glucosidase and α-amylase activity, which are enzymes responsible for breaking down starch and other substances into glucose during digestion [22]. Additionally, seaweed has been shown to reduce postprandial blood glucose levels in streptozotocin-induced diabetic mice [23]. Seaweeds have exhibited beneficial effects on blood glucose levels through various mechanisms. There is a need for a comprehensive evaluation of studies to establish the correlation between the consumption of brown algae and the prevention of diabetes. A previous meta-analysis examined the association between brown algae and blood glucose or glycolipid metabolism [24,25]. The current study aims to provide broader conclusion and analyze more research by incorporating recent studies.

## 2. Materials and Methods

### 2.1. Search Strategy

The PubMed, RISS, Google Scholar, ScienceDirect, and Cochrane Library databases were searched from inception to May 2023 to identify studies on the effects of algae supplementation on human blood glucose. The search keywords were as follows: (“brown algae” OR “brown seaweed” OR “Algae” OR “Seaweed”) AND (“blood glucose” OR diabetes OR “postprandial glucose” OR “hyperglycemia” OR “hypoglycemic” OR “antidiabetic”). Additionally, studies not covered by the previously mentioned search terms were manually searched using the scientific name of brown algae.

### 2.2. Selection Criteria

The inclusion criteria were developed in accordance with the PICO (Population, Intervention, Comparison, Outcome) principles and were as follows.

Population: Healthy participants, those with prediabetes, or participants with type 2 diabetes mellitusIntervention: Experimental studies exploring the effects of brown seaweeds or their extractsComparison: PlaceboOutcome: Randomized controlled trials (parallel or crossover)

### 2.3. Data Extraction, Quality Assessment, and Publication Bias

Two of the researchers reviewed the articles that passed the initial appraisal. The following data were extracted from eligible articles: (i) the primary author’s name; (ii) publication year; (iii) material used; (iv) RCT design; (v) subject description; (vi) participant count; (vii) duration; and (viii) relevant data for meta-analysis.

The risk of bias was assessed using the Cochrane Collaboration Risk of Bias tool. Five domains were evaluated: (i) randomization generation; (ii) deviations from the intended intervention; (iii) missing outcome data; (iv) measurement of the outcome; (v) selection of the reported result. The domains were evaluated using questions that were designed to gather information about features of the trial that are relevant to the risk of bias. For each study, the domains were judged as having a “low risk of bias”, a “high risk of bias”, or an “unclear risk of bias” [26].

### 2.4. Statistical Analysis

Comprehensive Meta-Analysis (CMA) software version 4.0 (Biostat Inc., Englewood, NJ, USA) was employed for all statistical analyses. Adjusted 95% confidence intervals (CIs) were provided to offer a quantitative estimation of the relationship between diabetes and brown seaweed and/or extract. The presence of statistical heterogeneity was assessed using the *I*^2^ index. A value below 25% indicates low heterogeneity, while a value exceeding 75% suggests high heterogeneity. To evaluate publication bias, funnel plots were generated through qualitative assessment, and the distribution was visually examined.

## 3. Results

### 3.1. Selection and Characteristics of Included Studies

The flowchart depicting the identification of relevant studies is presented in Figure 1. A total of 15,137 articles were identified through electronic searches of the following databases up to May 2023: PubMed (*n* = 108), RISS (*n* = 106), Google Scholar (*n* = 12,534), ScienceDirect (*n* = 2246), and the Cochrane Library (*n* = 143). A total of 15,104 studies were excluded based on the inclusion criteria. Subsequently, a thorough assessment of the full texts and the quality of the studies led to the exclusion of an additional 10 studies. Therefore, 23 papers were ultimately included for analysis (Figure 1).

The characteristics of the 23 included randomized controlled trials (RCTs) are succinctly outlined in Table 1. These studies encompassed individuals of varying genders and ages, with 7 studies adopting a crossover design and 16 studies adopting a parallel design. The intervention durations ranging from acute interventions to 180 days (Table 1).

### 3.2. Effect of Seaweed Intervention

#### 3.2.1. Fasting Blood Glucose and Fasting Blood Insulin Outcomes

Figure 2 illustrates the impact of brown algae or its extract on fasting blood glucose (FBG) and fasting blood insulin (FBI) [27,29,30,31,34,40,42,43,47,48]. The meta-analysis encompassed a total of 12 randomized controlled trials (RCTs) for FBG and 8 RCTs for FBI. The findings indicate that supplementation with algae did not yield a significant alteration in FBI levels (Figure 2b). Nevertheless, a noteworthy reduction in FBG levels was observed following seaweed consumption (Figure 2a: mean difference −0.165, 95% CI [−0.325, −0.005], *p* = 0.043, *I*^2^ = 1.714). Notably, FBG displayed low heterogeneity with an *I*^2^ of 1.714, whereas FBI exhibited high heterogeneity with an *I*^2^ of 91.837.

#### 3.2.2. Postprandial Blood Glucose

Figure 3 depicts the effects of brown algae or its extract on postprandial blood glucose levels at 60, 90, and 120 min [19,39,45,46]. The analysis encompassed a total of eight randomized controlled trials (RCTs) that examined postprandial blood glucose. Notably, significant reductions were observed at 60, 90, and 120 min following the consumption of seaweed or its extracts in comparison to the control group (Figure 3a: mean difference −0.738, 95% CI [−1.177, −0.298], *p* = 0.001, *I*^2^ = 64.825; Figure 3b: mean difference −0.729, 95% CI [−1.134, −0.325], *p* < 0.0001, *I*^2^ = 58.709; Figure 3c: mean difference −0.732, 95% CI [−1.295, −0.168], *p* = 0.011, *I*^2^ = 78.313). A moderate level of heterogeneity was observed at 60 and 90 min, while at 120 min, an *I*^2^ value of 78.313 indicated high levels of heterogeneity.

#### 3.2.3. HbA1c and HOMA-IR

Figure 4 illustrates the effects of brown algae or its extract on HbA1c and Homeostasis Model Assessment of Insulin Resistance (HOMA-IR) [27,29,34,35,36,40,42,47,48]. The analysis included a total of 10 randomized controlled trials (RCTs) reporting HbA1c and 7 RCTs, which also reported HOMA-IR. Consumption of seaweed supplements resulted in a significant reduction in both HbA1c and HOMA-IR levels compared to the control group, with a moderate level of heterogeneity, *I*^2^ = 26.309 and 26.721, respectively (Figure 4a: mean difference −0.278, 95% CI [−0.458, −0.099], *p* = 0.002, *I*^2^ = 26.309; Figure 4b: mean difference −0.263, 95% CI [−0.499, −0.027], *p* = 0.029, *I*^2^ = 26.721).

#### 3.2.4. Subgroup Analysis

Consumption of seaweed significantly reduced fasting blood glucose and HbA1c levels in the group with a ≥12-week duration of consumption. In terms of intake quantity, a noteworthy decrease in fasting blood glucose and postprandial blood glucose at 60, 90, and 120 min was observed in the group that consumed 1000 mg/day or more. Additionally, significant distinctions were observed based on the analysis method and study design. Noteworthy variations were found depending on the type of seaweed utilized. Specifically, for HbA1c, a substantial reduction was noted in the group that consumed both *Ascophyllum nodosum* and *Fucus vesiculosus* (95% CI [−0.433 (0.652, −0.233)], *p* = 0.002, *I*^2^ = 35.63). Moreover, significant reductions were observed in postprandial blood glucose levels at 90 and 120 min in the groups that consumed *Ecklonia cava*, *Laminaria digitata*, and *Undaria pinnatifida* in comparison to the control group (Table 2).

### 3.3. Publication Bias

The publication bias is illustrated test in the paper using a funnel plot and Egger’s test (Figure 5, Table 3). The calculated effect sizes exhibited a balanced distribution around the pooled effect sizes for fasting blood glucose, postprandial blood glucose 60~120 min, HbA1c, and HOMA-IR. In all outcomes except for fasting blood insulin, no significant publication bias was observed (fasting blood glucose: *p* = 0.939, fasting blood insulin: *p* = 0.007, postprandial blood glucose 60 min: *p* = 0.366, postprandial blood glucose 90 min: *p* = 0.560, postprandial blood glucose 120 min: *p* = 0.413, HbA1c: *p* = 0.366, HOMA-IR: *p* = 0.906).

## 4. Discussion

Examining the effect of seaweed consumption or its extracts on biomarkers associated with blood glucose regulation, a meta-analysis of 23 clinical studies was conducted. The results indicated that incorporating seaweed into the diet could potentially be effective in preventing and managing type 2 diabetes through improved blood sugar control. The meta-analysis of randomized controlled trials (RCTs) revealed that the group consuming seaweed showed decreased levels of postprandial blood glucose, HbA1c, and HOMA-IR compared to the control group. These indicators should be monitored sufficiently for diabetes management. For individuals with type 2 diabetes, impaired insulin secretion during the post-meal digestion process leads to postprandial hyperglycemia, which in turn triggers metabolic abnormalities in the liver or pancreas [49]. HbA1c is formed by the glycation of hemoglobin and leads to an average blood glucose level in an individual’s blood [50]. This serves as a standard of care (SOC) biomarker for diabetes monitoring. HOMA-IR is a key indicator for the primary prevention of diabetes, as it distinguishes early stage diabetes. It is calculated using the following formula and relies on fasting plasma insulin concentration: (HOMA-IR = fasting blood glucose × fasting insulin/constant) [51,52]. The decrease in postprandial blood glucose, HbA1c, and HOMA-IR as a result of consuming seaweed or its extracts indicates a reduced risk of diabetes patients experiencing post-meal hyperglycemia and signifies an improvement in blood sugar management and metabolic status.

In the subgroup analysis of this study, differences were observed based on factors such as duration, dosage, and scientific names. Concerning the duration, a significant reduction in HOMA-IR was noted in the group with a consumption period of less than 12 weeks, while fasting blood glucose and HbA1c significantly decreased in the group with a consumption period exceeding 12 weeks. Additionally, it was observed that the majority of biomarkers improved when the daily intake exceeded 1000 mg. These findings highlight the promising role of seaweed consumption in enhancing glycemic control.

Our research findings align with similar studies previously conducted by Kate Vaughan et al. (2022) and Ding K-x (2020). Ding K-x et al. reported significant reductions in fasting plasma glucose (mean difference −4.6, 95% CI [−7.9, −1.3], *I*^2^ = 99%) and postprandial plasma glucose (mean difference −7.1, 95% CI [−7.4, −6.9], *p* < 0.001, *I*^2^ = 99%) through seaweed consumption compared to the control group [25,53]. In contrast, Kate Vaughan et al. observed a trend of decreased fasting plasma glucose, postprandial 2 h glucose, and HOMA-IR with algae consumption. They also identified a significant reduction in HbA1c (mean difference −0.18%, 95% CI [−0.27, −0.10], *p* < 0.001, *I*^2^ = 35%) compared to the control group [25]. Unlike our study, Ding K-x’s assessment focused on plasma glucose levels, which tend to show slightly higher values due to the lower water content compared to whole blood. Nevertheless, directional changes in fasting and postprandial blood glucose were consistently observed [53]. Kate Vaughan’s study encompassed a broader analysis of various algal species, whereas our research specifically targeted brown seaweed for analysis, leading to distinctions in the types of studies included [25]. In the review by Animish and Jayasri (2023), a comparative analysis of antidiabetic research on three types of algae based on color revealed that research on brown algae is the most widely reported, ranging from in vitro studies to in vivo studies [54]. Thus, it is necessary to conduct a meta-analysis of research on brown algae for antidiabetic studies through research synthesis [55,56].

In this study, there were 17 studies that focused on monotherapy, surpassing the number of studies on complex interventions. Furthermore, in the subgroup analysis, both monotherapy and complex interventions exhibited significant improvements. Therefore, it is possible to anticipate a synergistic effect in complex interventions. However, due to the limited amount of data and its inherent constraints, additional research is necessary to investigate both single therapies and complex interventions.

The extracts from *Fucus vesiculosus* and *Ascophyllum nodosum* were assessed for their ability to inhibit α-amylase and α-glucosidase enzymes. The results indicated complete inhibition of α-amylase at a concentration of 30 μg/mL and α-glucosidase at a concentration of 2 μg/mL [57]. Additionally, when this extract was administered to mice fed a high-fat diet, there was a significant reduction in the average AUC (area under the curve) compared to the control group [57]. Glucagon-like peptide 1 (GLP-1 [7,8,9,10,11,12,13,14,15,16,17,18,19,20,21,22,23,24,25,26,27,28,29,30,31,32,33,34,35,36] amide or GLP-1) stimulates insulin secretion and synthesis while inhibiting glucagon release and is cleaved by dipeptidyl peptidase IV (DPP-IV) [58]. Therefore, the inhibition of DPP-IV has been studied as a novel therapeutic approach to decrease glucose production and increase insulin secretion, aiming to prevent postprandial hyperglycemia. In a study investigating the anti-hyperglycemic mechanism of the brown seaweed *Fucus vesiculosus* L. from the Barents Sea, the inhibition of Dipeptidyl Peptidase IV (DPP-IV) was assessed. These findings revealed that fucoidan extracted from *Fucus vesiculosus* L. demonstrated concentration-dependent inhibition of DPP-IV, with a maximum inhibition ranging from 60% to 75% within the concentration range of 0.02 to 200 μg/mL. This adds to the body of evidence endorsing the potential role of fucoidan extracted from brown algae in glucose regulation through inhibition [59].

*Ecklonia cava* is recognized for its high content of phlorotannins compared to other seaweeds, and the functional properties of its derivative, dieckol, have been extensively documented [60]. Administering dieckol to C57BL/KsJ-db/db mice resulted in a significant decrease in blood glucose and insulin levels compared to the control group. This effect is believed to be associated with the AMPK and Akt signaling pathways [60]. Phlorotannins, primarily found as secondary metabolites in brown seaweed, are polyphenolic derivatives extracted from the phloroglucinol unit [61,62]. Phlorotannins, including Dieckol and Eckol, have been reported to exhibit inhibitory effects on several enzymes, such as α-amylase, α-glucosidase, aldose reductase (AR), and angiotensin-converting enzyme (ACE). Additionally, human studies demonstrating postprandial blood glucose reduction effects through supplementation with seaweed from the Laminaria genus have also been reported [48]. Fucoxanthin, a carotenoid found in *Undaria pinnatifida*, exhibits notable inhibitory activity against α-glucosidase and effectively hinders the formation of advanced glycation end products (AGEs), suggesting its potential in regulating blood glucose levels [63]. Polyphenols were observed to inhibit α-glucosidase activity in a concentration-dependent manner in vitro (0.1, 0.19, 0.49, 0.98 mM: 25.43%, 56.74%, 84.14%, 90.97%). Furthermore, in diabetic mice, AUC was significantly reduced compared to that of the control group. Polyphenols have been reported to improve diabetes by enhancing glucose absorption, reducing insulin resistance, and preventing diabetic complications [48,64,65]. Alginate and fucoidan are types of polysaccharides naturally occurring in seaweeds, both known for their antidiabetic effects. Alginate notably reduced fasting blood glucose levels in male ICR mice with high-fat diet/streptozotocin-induced diabetes during the 2nd, 3rd, and 4th weeks compared to the control group. Fucoidan is reported to prevent hyperglycemia and increase insulin production as a complex sulfated polysaccharide [48].

In conclusion, this research has confirmed the effectiveness of consuming particular brown seaweeds, specifically *Ecklonia cava*, *Laminaria digitata*, and *Undaria pinnatifida*, in regulating postprandial blood glucose. Additionally, it was found that the beneficial effects become noticeable with a daily dosage exceeding 1000 mg and a consumption period of more than 12 weeks. The consumption of seaweed is on the rise in various countries, driven by an increasing global awareness of seaweed products, including those with certifications like halal food [66]. However, since seaweed is not universally integrated into diets worldwide, convincing the public of its benefits as a dietary supplement may pose challenges. Therefore, it appears that seaweed supplementation may be an effective strategy for blood sugar control, particularly in key consumer markets such as Asia and the United States. Additionally, safety assessment and management of potentially high levels of iodine, fiber, heavy metals, and contaminants that can be present in seaweed are essential considerations [67].

In our study, fasting blood insulin levels did not show statistical significance. In the case of fasting insulin, it was not statistically significant, likely due to the significant publication bias. Improving publication bias is essential, and additional research seems warranted.

The current study examines the influence of seaweed and its extracts as supplements on blood glucose levels. It is essential to assess the effects of red algae and green algae on blood glucose and to determine the differences in the effects of brown, red, and green algae. Additional RCTs are required to gain a precise understanding of the mechanisms involved. Moreover, the studies included in this research possess the strength of comprehensively screening the relationship between seaweed-derived substances and blood glucose levels. However, there has not been research conducted on the various extraction methods of seaweed. As the composition of extracts can vary depending on extraction methods, solvents, and other factors, further studies should focus on additional research through categorizing based on extraction methods.

## Figures and Tables

**Figure 1 nutrients-15-04987-f001:**
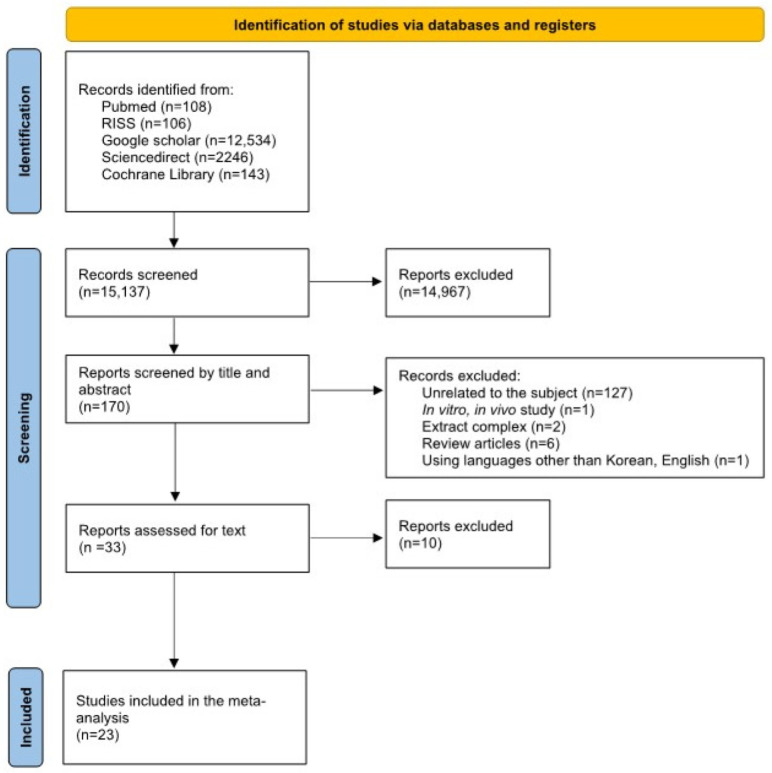
PRISMA flow diagram of studies included in the meta-analysis.

**Figure 2 nutrients-15-04987-f002:**
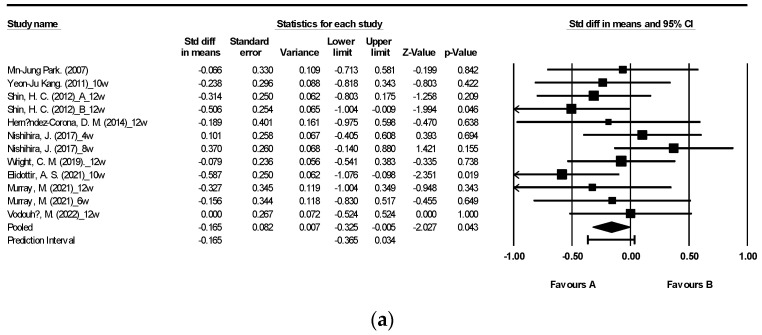
Forest plot of the effect of brown seaweeds or their extracts on fasting blood glucose (**a**) and fasting blood insulin (**b**) [27,29,30,31,34,40,42,43,47,48]. Each black square signifies a study’s point estimate and indicates its sample size—larger squares represent studies with more participants. The diamond shape represents the pooled mean difference.

**Figure 3 nutrients-15-04987-f003:**
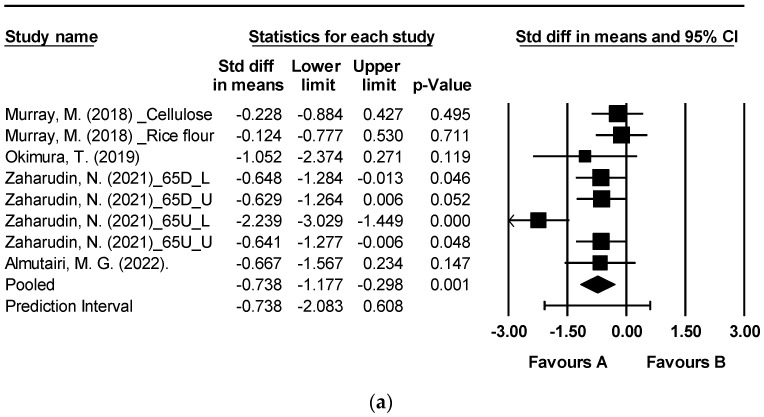
Forest plot of the effect of brown seaweeds or their extracts on postprandial blood glucose 60 min (**a**), postprandial blood glucose 90 min (**b**), and postprandial blood glucose 120 min (**c**) [19,39,45,46]. Each black square signifies a study’s point estimate and indicates its sample size—larger squares represent studies with more participants. The diamond shape represents the pooled mean difference.

**Figure 4 nutrients-15-04987-f004:**
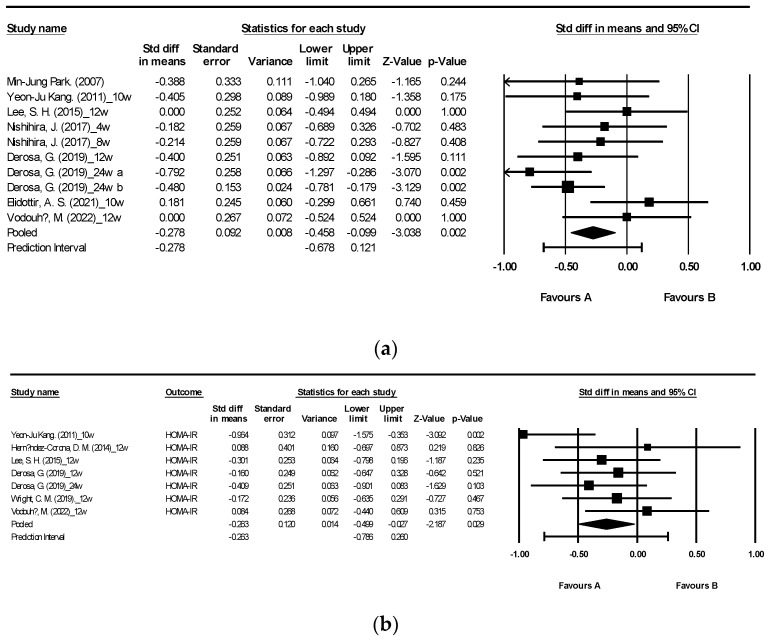
Forest plot of the effect of brown seaweeds or their extracts on HbA1c (**a**) and HOMA-IR (**b**) [27,29,34,35,36,40,42,47,48]. Each black square signifies a study’s point estimate and indicates its sample size—larger squares represent studies with more participants. The diamond shape represents the pooled mean difference.

**Figure 5 nutrients-15-04987-f005:**
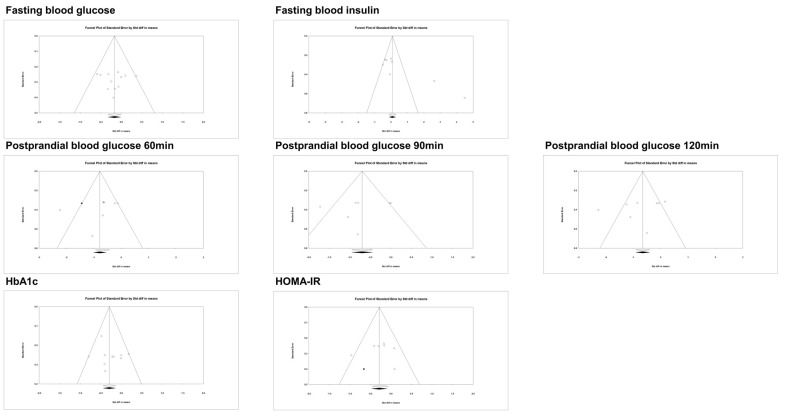
Funnel plots of publication bias.

**Table 1 nutrients-15-04987-t001:** Characteristics of included clinical studies.

Study Name	Scientific Name	Intervention	RCT Design	Diabetes	Country	Subject Number	Duration (Day)	Result
Min-Jung Park. [27]	*Laminaria Japonica*	*Laminaria Japonica* hot water extract 1400 mg	Parallel	Y	Republic of Korea	44	84	(-) FBG, HbA1c
Oh, J.K. [28]	*Ecklonia cava*	Drink contained 40 mg of *Ecklonia cava* Polyphenol per 100 mL (72 mg/day)	Cross-over	N	Republic of Korea	20	Acute	(-) Blood glucose level
Paradis, M.E. [18]	*Ascophyllum nodosum*, *Fucus vesiculosus*	InSea2^®^ (*Ascophyllum nodosum* and *Fucus vesiculosus* hot water extract powder) 500 mg	Cross-over	N	Danmark	23	Acute	(-) Plasma glucose iAUC, (↑) Plasma insulin iAUC
Kang, Y.-J. [29]	*Ishige okamurae*	Freeze-dried *Ishige okamurae* extract 1600 mg	Parallel	Y	Republic of Korea	60	70	(-) FBG, Insulin, HOMA-IR, (↓) HbA1c
Shin, H.C. [30]	*Ecklonia cava*	*Ecklonia cava* Polyphenol 72, 144 mg	Parallel	N	Republic of Korea	107	84	(↓) Glucose
Hernández-Corona, D.M. [31]	*-*	F-fucoidan (Green Foods) 500 mg	Parallel	N	NA	25	90	(-) Glucose, (↑) Insulin, HOMA-IR
Lee, S.H. [32]	*Ecklonia cava*	Dieckol-rich extract (AG-dieckol) from *E. cava* 500 mg	Parallel	N	Republic of Korea	80	84	(↓) Postprandial glucose, Insulin, (-) FPG, HbA1c, HOMA-IR
Choi, W.-c. [33]	*Laminaria Japonica*	γ-aminobutyric acid (GABA)-enriched fermented sea tangle 1000 mg	Parallel	N	Republic of Korea	21	56	(↑) IGF
Nishihira, J. [34]	*Kjellmaniella crassifolia Miyabe*	*Kjellmaniella crassifolia* Miyabe dietary fiber 800 mg	Parallel	N	Japan	60	56	(-) Glucose, HbA1c
Murray, M. [19]	*Fucus vesiculosus*	Maritech^®^ Synergy (*Fucus vesiculosus* powdered extract) 500, 2000 mg	Cross-over	N	Asian (Chinese, Indian, Indonesian), non-Asian (white Australian, British, Polish, Persian, Turkish, Italian, Greek)	39	Acute	(-) Blood glucose, Plasma insulin
Derosa, G. [35]	*Ascophyllum nodosum*, *Fucus vesiculosus*	Gdue^®^ (*Ascophyllum nodosum* and *Fucus vesiculosus* extract)	Parallel	Y	Caucasian patient	65	180	(↓) HbA1c, FPG, PPG, HOMA index, (-) FPI
Derosa, G. [36]	*Ascophyllum Nodosum*, *Fucus Vesiculosus*	InSea2^®^ (*Ascophyllum nodosum* and *Fucus vesiculosus* extract) 500 mg	Parallel	Y	Caucasian patient	175	180	(↓) FPG, PPG, HbA1c
Murray, M. [37]	*Fucus Vesiculosus*	*Fucus Vesiculosus* powder extract 2000 mg	Cross-over	N	Melbourne, Australia	23	Acute	(-) Blood glucose, Plasma insulin
Nishimura, M. [38]	*Laminariaceae*	*Laminariaceae powder* 2000 mg	Parallel	N	Japan	70	42	(↓) FPG, (-) Insulin, HbA1c, HOMA-IR
Okimura, T. [39]	*Ascophyllum nodosum*	Ascophyllan HS (*Ascophyllum nodosum* water extract) 100 mg	Parallel	N	Japan	NA	56	(-) Blood glucose, Serum HbA1c, (↓) Fluctuation of HbA1c
Wright, C.M. [40]	*Fucus vesiculosus*	Maritech^®^ Synergy (*Fucus vesiculosus* fucoidan/polyphenol extract) 1000 mg	Parallel	N	Australian New Zealand	72	90	(-) Insulin, HOMA-IR, FBG
van den Driessche, J.J. [41]	*Undaria pinnatifida*	*Undaria pinnatifida* 4800 mg	Cross-over	N	The Netherlands	36	17	(-) Glucose
Elidottir, A.S. [42]	*Fucus vesiculosus*	*Fucus Vesiculosus* extract 1200 mg	Parallel	N	Iceland	76	70	(↓) Glucose, Insulin, (-) HbA1c
Murray, M. [43]	*Fucus vesiculosus*	Maritech^®^ Synergy (*Fucus vesiculosus* powdered extract) 2000 mg	Parallel	N	Asian, non-Asian (Caucasian, African, Poly-ethnic)	38	96	(-) Glucose, Insulin
van den Driessche, J.J. [44]	*Undaria pinnatifida*	*Undaria pinnatifida* 4800 mg	Cross-over	N	The Netherlands	52	17	(-) Glucose
Zaharudin, N. [45]	*Laminaria digitata*, *Undaria pinnatifida*	Seaweed salads (whole or chopped leafs) 5000 mg	Cross-over	N	NA	20	Acute	(↓) Glucose, Insulin, C-peptide, GLPx1-1
Almutairi, M.G. [46]	*Ecklonia cava*	Seanol (*Ecklonia cava* extract) 600 mg	Parallel	Y	Saudi Arabia	20	Acute	(-) FBG, PBG 30 min, 60 min, Peak concentration, FBI, PBI, (↓) PBG 90 min, 120 min
Vodouhè, M. [47]	*Ascophyllum nodosum*, *Fucus vesiculosus*	InSea2^®^ (*Ascophyllum nodosum* and *Fucus vesiculosus* extract) 500 mg	Parallel	Y	NA	56	84	(-) FBG, FBI, C-peptide, HbA1c

**Table 2 nutrients-15-04987-t002:** Results of subgroup analysis.

Outcome	Subgroup	No. of Trials	Effect Size (95% CI)	*p*-Value	*I*^2^ (%)
Fasting blood glucose	Duration				
	<12 weeks	5	−0.099 (−0.442, 0.244)	0.571	49.441
	≥12 weeks	7	−0.216 (−0.425, −0.006)	0.044 *	0
	Dose & Intake direction				
	<1000 mg/day	6	−0.099 (−0.357, 0.158)	0.450	30.716
	≥1000 mg/day	5	−0.314 (−0.582, −0.046)	0.022 *	0
	Intervention				
	Water extract	2	−0.026 (−0.433, 0.381)	0.900	0
	Powder	4	−0.177 (−0.466, 0.111)	0.228	0
	Polyphenol	2	−0.408 (−0.757, −0.060)	0.022 *	0
	Fucoidan	1	−0.189 (−0.975 0.598)	0.638	0
	Dietary fiber	2	0.235 (−0.125, 0.594)	0.201	0
Fasting blood insulin	Intervention				
	Water extract	1	0.048 (−0.476, 0.572)	0.857	0
	Powder	4	1.578 (−0.208, 3.365)	0.083	95.993
	Fucoidan	1	−0.062 (−0.847, 0.722)	0.876	0
	Dieckol	1	−0.265 (−0.761, 0.231)	0.295	0
HbA1c	Scientific name				
	*Ascophyllum nodosum*, *Fucus vesiculosus*	4	−0.433 (0.652, −0.233)	0.002 *	35.63
	*Ecklonia cava*	1	0.000 (−0.494, 0.494)	1.000	0
	*Fucus vesiculosus*	1	0.181 (−0.299, 0.661)	0.459	0
	*Ishige okamurae*	1	−0.405 (−0.989, 0.180)	0.175	0
	*Kjellmaniella crassifolia Miyabe*	2	−0.198 (0.557, 0.161)	0.280	0
	*Laminaria Japonica*	1	−0.388 (−1.040, 0.265)	0.244	0
	Diabetes				
	Y	6	−0.434 (−0.623, −0.245)	0.000 *	0
	N	4	−0.046 (−0.295, −0.202)	0.715	0
	Duration				
	<12 weeks	4	−0.129 (−0.387, 0.129)	0.328	0
	≥12 weeks	6	−0.360 (−0.593, −0.128)	0.002 *	31.245
	Intervention				
	Water extract	5	−0.432 (−0.655, 0.210)	<0.0001 *	14.617
	Powder	1	−0.405 (−0.989, 0.180)	0.175	0
	Dietary fiber	2	−0.198 (−0.557, 0.161)	0.280	0
	Dieckol	1	0.000 (−0.494, 0.494)	1.000	0
HOMA-IR	Duration				
	<12 weeks	1	−0.964 (−1.575, −0.353)	0.002 *	0
	≥12 weeks	6	−0.176 (−0.388, 0.353)	0.103	0
	Intervention				
	Water extract	3	−0.172 (−0.461, 0.117)	0.244	0
	Powder	2	−0.542 (−1.316, 0.233)	0.170	75.614
	Fucoidan	1	0.088 (−0.697, 0.873)	0.826	0
	Dieckol	1	−0.301 (−0.798, 0.196)	0.235	0
Postprandial blood glucose (60 min)	Design				
	Cross-over	6	−0.725 (−1.262, −0.188)	<0.0001 *	74.476
	Parallel	2	−0.789 (−1.533, −0.044)	0.038 *	0
	Dose & Intake direction				
	<1000 mg/day	4	−0.347 (−0.740, 0.046)	0.084	0
	≥1000 mg/day	4	−1.007 (−1.703, −0.312)	0.005 *	76.814
	Intervention				
	Water extract	1	−1.052 (−2.374, 0.271)	0.119	0
	Powder	2	−0.176 (−0.639, 0.287)	0.457	0
	Polyphenol	1	−0.667 (−1.567, 0.234)	0.147	0
	Fresh seaweed	4	−1.007 (−1.703, −0.312)	0.005 *	76.814
Postprandial blood glucose (90 min)	Scientific name				
	*Ascophyllum nodosum*	1	−0.805 (−2.094, 0.484)	0.221	0
	*Ecklonia cava*	1	−1.044 (−1.978, 0.110)	0.029 *	0
	*Fucus Vesiculosus*	2	−0.014 (−0.476, 0.447)	0.951	0
	*Laminaria digitata*, *Undaria pinnatifida*	4	−1.015 (−1.432, −0.597)	<0.0001 *	36.879
	Design				
	Cross-over	6	−0.667 (−0.936, −0.397)	<0.0001 *	69.414
	Parallel	2	0.962 (−1.718, −0.205)	0.013 *	0
	Diabetes				
	Y	7	−0.672 (−0.936, −0.409)	<0.0001 *	63.392
	N	1	1.044 (−1.978, −0.110)	0.029 *	0
	Dose & Intake direction				
	<1000 mg/day	4	−0.330 (−0.831, 0.172)	0.198	32.879
	≥1000 mg/day	4	−1.015 (−1.432, −0.597)	<0.0001 *	32.991
	Duration				
	Acute	7	−0.726 (−1.164, −0.288)	0.001 *	64.552
	Long	1	−0.805 (−2.094, 0.484)	0.221	0
	Intervention				
	Water extract	1	−0.805 (−2.094, 0.484)	0.221	0
	Powder	2	−0.014 (−0.476, 0.447)	0.951	0
	Polyphenol	1	−1.044 (−1.978, −0.110)	0.029 *	0
	Fresh seaweed	4	−1.015 (−1.432, −0.597)	<0.0001 *	36.879
Postprandial blood glucose (120 min)	Scientific name				
	*Ascophyllum nodosum*	1	−0.500 (−1.758, 0.759)	0.437	0
	*Ecklonia cava*	1	−1.101 (−2.041, −0.160)	0.022 *	0
	*Fucus Vesiculosus*	2	−0.097 (−0.559, 0.365)	0.682	0
	*Laminaria digitata*, *Undaria pinnatifida*	4	−1.027 (−1.982, −0.071)	0.035 *	87.332
	Design				
	Cross-over	6	−0.710 (−1.395, −0.025)	0.042 *	84.025
	Parallel	2	−0.885 (−1.639, −0.132)	0.021 *	0
	Diabetes				
	Y	7	−0.686 (−1.309, −0.064)	0.031 *	80.851
	N	1	−1.101 (−2.041, 0.022)	0.022 *	0
	Dose & Intake direction				
	<1000 mg/day	4	−0.336 (−0.781, 0.109)	0.139	17.631
	≥1000 mg/day	4	−1.027 (−1.982, −0.071)	0.035 *	87.332
	Duration				
	Acute	7	−0.757 (−1.370, 0.144)	0.015 *	81.377
	Long	1	−0.5 (−1.758, 0.759)	0.437	0
	Intervention				
	Water extract	1	−0.500 (−1.758, 0.759)	0.437	0
	Powder	2	−0.097 (−0.559, 0.365)	0.682	0
	Polyphenol	1	−1.101 (−2.041, −0.160)	0.022 *	0
	Fresh seaweed	4	−1.027 (−1.982, −0.071)	0.035 *	87.332

* *p* < 0.05, compared with placebo.

**Table 3 nutrients-15-04987-t003:** Results of Egger’s test for publication bias.

Outcome	95% Lower Limit	95% Upper Limit	t-Value	*df*	*p*-Value
Fasting blood glucose	−4.46	4.52	0.01	10	0.989
Fasting blood insulin	3.79	15.95	3.97	6	0.007
Postprandial blood glucose 60 min	−10.17	4.36	0.98	6	0.366
Postprandial blood glucose 90 min	−9.05	5.40	0.62	6	0.560
Postprandial blood glucose 120 min	−13.06	6.16	0.88	6	0.413
HbA1c	−2.64	5.42	0.79	8	0.449
HOMA-IR	−8.58	7.79	0.12	5	0.906

Abbreviations: *df*, Degrees of freedom.

## Data Availability

The data presented in this study are available in insert article.

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
