# Peer review of "Brown Seaweed Consumption as a Promising Strategy for Blood Glucose Management: A Comprehensive Meta-Analysis"

_nutrients, 2023, doi:10.3390/nu15234987_

Round 1

Reviewer 1 Report

Comments and Suggestions for Authors

I have read the manuscript and have several comments and recommendations.
1. The authors studied a large amount of literature on the use of algae for diabetes. However, it should be noted that the mechanism of antidiabetic action involves only inhibition of α-glucosidase and α-amylase activities. In recent years, numerous studies have proven that inhibition of the enzyme dipeptidyl petidase-IV is one of the mechanisms of the antidiabetic action of modern drugs. The literature contains data on mechanisms of bioactivities of fucoidan from the brown seaweed Fucus vesiculosus L. of the Barents sea. The authors showed that the inhibition of dipeptidyl peptidase-IV is one of the possible mechanisms involved in the anti-hyperglycemic activity of fucoidan. Include additional information about the different mechanisms of antidiabetic action of algae preparations/extracts.
2. Much attention has been paid to the study of algae components. The authors review clinical trial data. What are the results of these numerous clinical trials? Are there algae preparations recommended as medicines for reducing sugar, i.e. antidiabetic drugs?
3. It is necessary to discuss the question of what may hinder the introduction of algae preparation into medical practice to reduce sugar.
4. How do the authors see the prospects for using algae preparations: as monotherapy or complex?

Author Response

List of changes

Manuscript ID: nutrients-2702615

Title: Brown Seaweed Consumption as a Promising Strategy for Blood Glucose Management: A Comprehensive Meta-Analysis

Dear Editor,

Thank you very much for your consideration of our manuscript and request for a revised version. We have copy and pasted all reviewers’ comments and address each one individually. As you will see, we made a number of changes in our manuscript to incorporate the questions and suggestions by the reviewers as thoroughly as possible.  Below is a list of the changes following the order of the comments.

Reviewer #1

  1. The authors studied a large amount of literature on the use of algae for diabetes. However, it should be noted that the mechanism of antidiabetic action involves only inhibition of α-glucosidase and α-amylase activities. In recent years, numerous studies have proven that inhibition of the enzyme dipeptidyl petidase-IV is one of the mechanisms of the antidiabetic action of modern drugs. The literature contains data on mechanisms of bioactivities of fucoidan from the brown seaweed Fucus vesiculosus L. of the Barents sea. The authors showed that the inhibition of dipeptidyl peptidase-IV is one of the possible mechanisms involved in the anti-hyperglycemic activity of fucoidan. Include additional information about the different mechanisms of antidiabetic action of algae preparations/extracts.

→ We appreciate the reviewer for the valuable comment. We have discussed additional information about the various mechanisms of antidiabetic action of algae preparations/extracts (Lines 269~280).

  1. Much attention has been paid to the study of algae components. The authors review clinical trial data. What are the results of these numerous clinical trials? Are there algae preparations recommended as medicines for reducing sugar, i.e. antidiabetic drugs?

→ We thank the reviewer for this important comment. We have decided to add the results of these numerous clinical trials by including a 'Result' column (Table 1).

→  Brown algae have been confirmed to be available as patented and commercially marketed dietary supplements. InSea2® is one such dietary supplement, composed of extracts from the brown seaweeds Ascophyllum nodosum and Fucus vesiculosus, which contain phlorotannins. It has been approved by Health Canada for its two primary benefits: helping maintain healthy blood glucose levels and reducing the glycemic index of ingested foods (NPN 80033840) (Line 59~63).

  1. It is necessary to discuss the question of what may hinder the introduction of algae preparation into medical practice to reduce sugar.

→  Thank you for the comments. The safety assessment and management of potentially high levels of iodine, fiber, heavy metals, and contaminants that may be present in seaweed are essential factors to consider. Additionally, as suggested by other reviewers, it is important to take into account the cultural and dietary acceptance of seaweed, given that it is not widely consumed globally but is predominantly consumed in Asian countries (Line 315~317).

  1. How do the authors see the prospects for using algae preparations: as monotherapy or complex?

→   Thank you for the comments. We observed that among the references used in this meta-analysis, there were 17 studies on monotherapy, which outnumbered the studies on complex interventions. Furthermore, in the sub-analysis, both monotherapy and complex interventions demonstrated significant improvements. Therefore, we have chosen to include a statement in the Discussion section, highlighting the necessity for additional studies comparing complex interventions and monotherapy (Line 258~263).

Reviewer 2 Report

Comments and Suggestions for Authors

Limitations:

I have a problem with authors conclusion stating that “seaweed supplementation appears to be a promising strategy for reducing postprandial blood glucose…”. Because to be effective strategy, they need first to show that seaweed supplementation is a well-accepted practice globally. In some countries it is an integrated part of their food habits, while in most countries it is not. Authors may elaborate on their conclusion in a more specific manner.

Minor suggestions:

Lines 42-44: “Approximately 30% of individuals using metformin experience gastrointestinal disturbances and mineral deficiencies, such as vitamin B12 and folic acid deficiency”. Note that vitamin B12 and folic are vitamins not minerals.

Line 104: cite a reference.

For “A value below 25% indicates low heterogeneity, while a value exceeding 75% suggests high heterogeneity” please cite a reference.

Author Response

                List of changes

Manuscript ID: nutrients-2702615

Title: Brown Seaweed Consumption as a Promising Strategy for Blood Glucose Management: A Comprehensive Meta-Analysis

Dear Editor,

Thank you very much for your consideration of our manuscript and request for a revised version. We have copy and pasted all reviewers’ comments and address each one individually. As you will see, we made a number of changes in our manuscript to incorporate the questions and suggestions by the reviewers as thoroughly as possible.  Below is a list of the changes following the order of the comments.

Reviewer #2: Limitations

  1. I have a problem with authors conclusion stating that “seaweed supplementation appears to be a promising strategy for reducing postprandial blood glucose…”. Because to be effective strategy, they need first to show that seaweed supplementation is a well-accepted practice globally. In some countries it is an integrated part of their food habits, while in most countries it is not. Authors may elaborate on their conclusion in a more specific manner.

 We thank the reviewer for this important comment. We have added a more specific conclusion (Line 310~315).

< Minor suggestions>

  1. Lines 42-44: “Approximately 30% of individuals using metformin experience gastrointestinal disturbances and mineral deficiencies, such as vitamin B12 and folic acid deficiency”. Note that vitamin B12 and folic are vitamins not minerals.

  Thank you for the comments. We have replaced the term ‘mineral’ with the word ‘vitamin’ (Lines 43).

  1. Line 104: cite a reference. For “A value below 25% indicates low heterogeneity, while a value exceeding 75% suggests high heterogeneity” please cite a reference.

  Thank you for the comments. We have added the reference (Lines 107).

Round 2

Reviewer 1 Report

Comments and Suggestions for Authors

The authors made the necessary corrections. There are no more questions for me. I recommend the manuscript for publication in Nutrients.

Author Response

 We appreciate the reviewer for your precious time in reviewing our paper and providing valuable comments.